# Scaling Transformers for Skillful and Reliable Medium-range Weather Forecasting

**Tung Nguyen**
UCLA

**Rohan Shah**
UCLA, CMU

**Hritik Bansal**
UCLA

**Troy Arcomano**
Argonne National Laboratory

**Romit Maulik**
Argonne National Laboratory, Penn State University

**Veerabhadra Kotamarthi**
Argonne National Laboratory

**Ian Foster**
Argonne National Laboratory

**Sandeep Madireddy**
Argonne National Laboratory

**Aditya Grover**
UCLA

## Abstract

Weather forecasting is a fundamental problem for anticipating and mitigating the impacts of climate change. Recently, data-driven approaches for weather forecasting based on deep learning have shown great promise, achieving accuracies that are competitive with operational systems. However, those methods often employ complex, customized architectures without sufficient ablation analysis, making it difficult to understand what truly contributes to their success. Here we introduce Stormer, a simple transformer model that achieves state-of-the-art performance on weather forecasting with minimal changes to the standard transformer backbone. We identify the key components of Stormer through careful empirical analyses, including weather-specific embedding, randomized dynamics forecast, and pressure-weighted loss. At the core of Stormer is a randomized forecasting objective that trains the model to forecast the weather dynamics over varying time intervals. During inference, this allows us to produce multiple forecasts for a target lead time and combine them to obtain better forecast accuracy. On Weather-Bench 2, Stormer performs competitively at short to medium-range forecasts and outperforms current methods beyond 7 days, while requiring orders-of-magnitude less training data and compute. Additionally, we demonstrate Stormer's favorable scaling properties, showing consistent improvements in forecast accuracy with increases in model size and training tokens.

## 1 Methodology

We introduce Stormer, an effective deep learning model for weather forecasting. We focus on the simplicity of the architecture, and aim to show that such an architecture can achieve competitive forecast performances with a well-designed framework. We first present the overall training and inference procedure of Stormer, and proceed to describe the model architecture we implement in practice. We provide an in-depth discussion about the different design choices of Stormer in Appendix D and empirically demonstrate the importance of each component in Appendix F.2.

### 1.1 Training

We train Stormer to forecast the weather dynamics $\Delta_{\delta t} = X_{\delta t} - X_0$, which is the difference between two consecutive weather conditions, $X_0$ and $X_{\delta t}$, across the time interval $\delta t$. A common practice in previous works (Keisler, 2022; Lam et al., 2023) is to use a small fixed value of $\delta t$ such as 6 hours. However, as we show in Figure 6a, while small intervals tend to work well for short lead times, larger intervals excel at longer lead times (beyond 7 days) due to less error accumulation. Therefore, having a model that can produce forecasts at different intervals and combine them in an effective manner has the potential to improve the performance of single-interval models. This motivates our *randomized dynamics forecasting objective*, which trains Stormer to forecast the dynamics at

random intervals $\delta t$ by conditioning on $\delta t$:

$$\mathcal{L}(\theta) = \mathbb{E}_{\delta t \sim P(\delta t), (X_0, X_{\delta t}) \sim \mathcal{D}} \left[ ||f_\theta(X_0, \delta t) - \Delta_{\delta t}||_2^2 \right], \tag{1}$$

in which $P(\delta t)$ is the distribution of the random interval. In our experiments, unless otherwise specified, $P(\delta t)$ is a uniform distribution over three values $\delta t \sim \mathcal{U}\{6, 12, 24\}$. These three time intervals play an important role in atmospheric dynamics. The 6 and 12-hour values help to encourage the model to learn and resolve the diurnal cycle (day-night cycle), one of the most important oscillations in the atmosphere driving short-term dynamics (e.g., temperature over the course of a day). The 24-hour value filters the effects of the diurnal cycle and allows the model to learn longer, synoptic-scale dynamics which are particularly important for medium-range weather forecasting (Holton, 2004).

### 1.1.1 PRESSURE-WEIGHTED LOSS

Due to the large number of variables being predicted, we use a physics-based weighting function to weigh variables near the surface higher. Since each variable lies on a specific pressure level, we can use pressure as a proxy for the density of the atmosphere at each level. This weighting allows the model to prioritize near-surface variables, which are important for weather forecasting and have the most societal impact. The final objective function that we use for training is:

$$\mathcal{L}(\theta) = \mathbb{E} \left[ \frac{1}{VHW} \sum_{v=1}^{V} \sum_{i=1}^{H} \sum_{j=1}^{W} w(v) L(i) (\widehat{\Delta}_{\delta t}^{vij} - \Delta_{\delta t}^{vij})^2 \right]. \tag{2}$$

The expectation is over $\delta t$, $X_0$, and $X_{\delta t}$ which we omit for notational simplicity. In this equation, $w(v)$ is the weight of variable $v$, and $L(i)$ is the latitude-weighting factor.

### 1.1.2 MULTI-STEP FINETUNING

To produce forecasts at a lead time beyond the training intervals, we roll out the model several times. Since the model's forecasts are fed back as input, the forecast error accumulates as we roll out more steps. To alleviate this issue, we finetune the model on a multi-step loss function. Specifically, for each gradient step, we roll out the model $K$ times, and average the objective (2) over the $K$ steps. The multi-step loss is thus:

$$\mathcal{L}(\theta) = \mathbb{E} \left[ \frac{1}{KVHW} \sum_{k=1}^{K} \sum_{v=1}^{V} \sum_{i=1}^{H} \sum_{j=1}^{W} w(v) L(i) (\widehat{\Delta}_{k\delta t}^{vij} - \Delta_{k\delta t}^{vij})^2 \right]. \tag{3}$$

In practice, we implement a three-phase training procedure for Stormer. In the first phase, we train the model to perform single step forecasting, which is equivalent to optimizing the objective in (2). In the second and third phases, we finetune the trained model from the preceding phase with $K = 4$ and $K = 8$, respectively. We use the same sampled value of the interval $\delta t$ for all $K$ steps. We tried randomizing $\delta t$ at each rollout step, but found that doing so destabilized training as the loss value at each step is of different magnitudes, hurting the final performance of the model.

## 1.2 INFERENCE

At test time, Stormer can produce forecasts at any time interval $\delta t$ used during training. Thus the model can generate multiple forecasts for a target lead time $T$ by creating different combinations of $\delta t$ that sum to $T$. We consider two inference strategies for generating forecasts:

**Homogeneous** We only consider homogeneous combinations of $\delta t$, i.e., combinations with just one value of $\delta t$. For example, for $T = 24$ we consider [6, 6, 6, 6], [12, 12], and [24].

**Best *m* in *n*** We generate $n$ different, possibly heterogeneous combinations of $\delta t$, evaluate each combination on the validation set, and pick $m$ combinations with the lowest losses for testing.

## 1.3 MODEL ARCHITECTURE

We instantiate the framework presented in Section 1.1 with a simple Transformer (Vaswani et al., 2017)-based architecture. Due to the similarity of weather forecasting to various dense prediction tasks in computer vision, one might consider applying Vision Transformer (ViT) (Dosovitskiy et al., 2020) for this task. However, weather data is distinct from natural images, primarily due to its significantly higher number of input channels. These channels represent atmospheric variables with intricate physical relationships. For example, the wind fields in the atmosphere are closely related to the gradient and shape of the geopotential field, while the wind field redistributes moisture and heat around the globe. Effectively modeling these interactions is critical to forecast accuracy.

## 1.4 WEATHER-SPECIFIC EMBEDDING

The standard patch embedding module in ViT, which uses a linear layer for embedding all input channels within a patch into a vector, may not sufficiently capture the complex interactions among input atmospheric variables. Therefore, we adopt for our architecture a weather-specific embedding module, consisting of two components, *variable tokenization* and *variable aggregation*.

**Variable tokenization** Given an input of shape $V \times H \times W$, variable tokenization linearly embeds each variable independently to a sequence of shape $(H/p) \times (W/p) \times D$, in which $p$ is the patch size and $D$ is the hidden dimension. We then concatenate the output of all variables, resulting in a sequence of shape $(H/p) \times (W/p) \times V \times D$.

**Variable aggregation** We employ a single-layer cross-attention mechanism with a learnable query vector to aggregate information across variables. This module operates over the variable dimension on the output from the tokenization stage to produce a sequence of shape $(H/p) \times (W/p) \times D$ which is then fed to the transformer backbone. This module offers two primary advantages. First, it reduces the sequence length by a factor of $V$, significantly alleviating the computational cost as we use transformer to process the sequence. Second, unlike standard patch embedding, the cross-attention layer allows the models to learn non-linear relationships among input variables, enhancing the model's capacity to capture complex physical interactions. We present the complete implementation details of the weather-specific embedding in Appendix E.

### 1.4.1 STORMER TRANSFORMER BLOCK

Following weather-specific embedding, the tokens are processed by a stack of transformer blocks (Vaswani et al., 2017). In addition to the input $X_0$, the block also needs to process the time interval $\delta t$. We do this by replacing the standard layer normalization module used in transformer blocks with adaptive layer normalization (adaLN) (Perez et al., 2018). In adaLN, instead of learning the scale and shift parameters $\gamma$ and $\beta$ as independent parameters of the network, we regress them with an one-layer MLP from the embedding of $\delta t$. Compared to ClimaX (Nguyen et al., 2023) which only adds the lead time embedding to the tokens before the first attention layer, adaLN is applied to every transformer block, thus amplifying the conditioning signal. Figure 6c shows the consistent improvement of adaLN over the additive lead time embedding used in ClimaX. Adaptive layer norm was widely used in both GANs (Karras et al., 2019; Brock et al., 2018) and Diffusion (Dhariwal & Nichol, 2021; Peebles & Xie, 2023) to condition on additional inputs such as time steps or class labels. Figure 3 illustrates Stormer's architecture. We refer to (Nguyen et al., 2023) for illustrations of the weather-specific embedding block.

## 2 EXPERIMENTS

We compare Stormer withstate-of-the-art weather forecasting methods, and conduct extensive ablation analyses to understand the importance of each component in Stormer. We also study Stormer scalability by varying model size and the number of training tokens. We conduct all experiments on WeatherBench 2 (WB2) (Rasp et al., 2023), a standard benchmark that provides training data, a set of state-of-the-art models, and an evaluation framework for comparing data-driven weather forecasting methods. In the following results, unless specified otherwise, we use the same training and evaluation setup for Stormer. We provide complete experiment details in Appendix E.

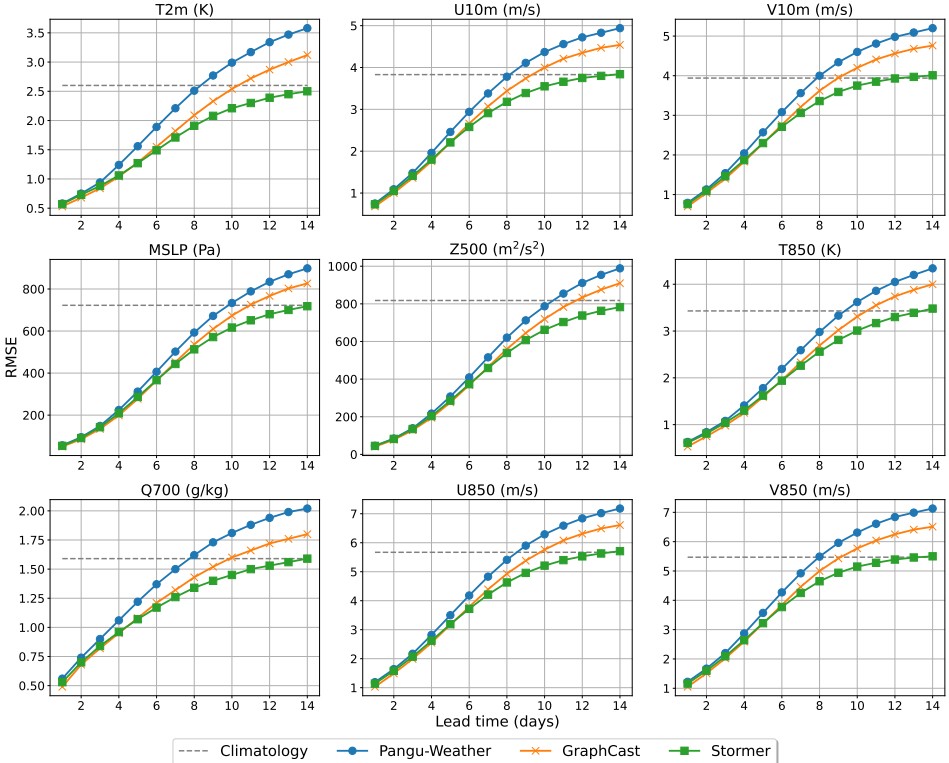

Figure 1: Global forecast verification results of Stormer and the baselines. We show the latitude-weighted RMSE for select variables. Stormer is on par or outperforms each of the benchmark models for the shown variables. During the later portion of the forecasts, Stormer gains $\sim 1$ day of forecast skill with respect to climatology compared to the next best deep learning model. We note that Stormer was trained on much lower resolution data ($1.40625°$) compared to Pangu-Weather ($0.25°$) and GraphCast ($0.25°$).

**Data:** We train and evaluate Stormer on the ERA5 dataset from WB2, which is the curated version of the ERA5 reanalysis data provided by the European Center for Medium-Range Weather Forecasting (ECMWF) (Hersbach et al., 2020). In its raw form, ERA5 contains hourly data from 1979 to the current time at $0.25°$ ($721\times1440$ grids) resolution, with different atmospheric variables spanning 137 pressure levels plus the Earth's surface. WB2 downsamples this data to 6-hourly with 13 pressure levels and provides different spatial resolutions. In this work, we use the $1.40625°$ ($128\times256$ grids) data. We use four surface-level variables – 2-meter temperature (T2m), 10-meter U and V components of wind (U10 and V10), and Mean sea-level pressure (MSLP), and five atmospheric variables – Geopotential (Z), Temperature (T), U and V components of wind (U and V), and Specific humidity (Q), each at 13 pressure levels $\{50, 100, 150, 200, 250, 300, 400, 500, 600, 700, 850, 925, 1000\}$. We train Stormer on data from 1979 to 2018, validate in 2019, and test in 2020, which is the common year for testing in WB2.

**Results:** Figure 1 evaluates different methods on forecasting nine key weather variables at lead times from 1 to 14 days. For short-range, 1–5 day forecasts, Stormer's accuracy is on par with or exceeds that of Pangu-Weather, but lags slightly behind GraphCast. *At longer lead times, Stormer excels, consistently outperforming both Pangu-Weather and GraphCast from day 6 onwards by a large margin*. Moreover, the performance gap increases as we increase the lead time. At 14 day forecasts, Stormer performs better than GraphCast by $10\%-20\%$ across all 9 key variables. Stormer is also the only model in this comparison that performs better than Climatology at long lead times, while other methods approach or even do worse than this simple baseline. The model's superior performance at long lead times is attributed to the use of randomized dynamics training, which improves forecast accuracy by averaging out multiple forecasts, especially when individual forecasts begin to diverge.

Moreover, we also note that Stormer achieves this performance with much less compute and training data compared to the two deep learning baselines. We train Stormer on 6-hourly data of $1.40625°$ with 13 pressure levels, which is approximately $190\times$ less data than Pangu-Weather's hourly data at $0.25°$ and $90\times$ less than that used for GraphCast, which also uses 6-hourly data but at a $0.25°$ resolution with 37 pressure levels. The training of Stormer was completed in under 24 hours on 128 A100 GPUs. In contrast, Pangu-Weather took 60 days to train four models on 192 V100 GPUs, and GraphCast required 28 days on 32 TPUv4 devices. This training efficiency will facilitate future works that build upon our proposed framework.

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

## A  INTRODUCTION

Weather forecasting is a fundamental problem for science and society. With increasing concerns about climate change, accurate weather forecasting helps prepare and recover from the effects of natural disasters and extreme weather events, while serving as an important tool for researchers to better understand the atmosphere. Traditionally, atmospheric scientists have relied on numerical weather prediction (NWP) models (Bauer et al., 2015). These models utilize systems of differential equations describing fluid flow and thermodynamics, which can be integrated over time to obtain future forecasts (Lynch, 2008; Bauer et al., 2015). Despite their widespread use, NWP models suffer from many challenges, such as parameterization errors of important small-scale physical processes, including cloud physics and radiation (Stensrud, 2009). Numerical methods also incur high computation costs due to the complexity of integrating a large system of differential equations, especially when modeling at fine spatial and temporal resolutions. Furthermore, NWP forecast accuracy does not improve with more data, as the models rely on the expertise of climate scientists to refine equations, parameterizations, and algorithms (Magnusson & Källén, 2013).

To address the challenges of NWP models, there has been an increasing interest in data-driven approaches based on deep learning for weather forecasting (Dueben & Bauer, 2018; Scher, 2018; Weyn et al., 2019). The key idea involves training deep neural networks to predict future weather conditions using historical data, such as the ERA5 reanalysis dataset (Hersbach et al., 2018; 2020; Rasp et al., 2020; 2023). Once trained, these models can produce forecasts in a few seconds, as opposed to the hours required by typical NWP models. Because of the similar spatial structure between weather data and natural images, early works in this space attempted to adopt standard vision architectures such as ResNet (Rasp & Thuerey, 2021; Clare et al., 2021) and UNet (Weyn et al., 2020) for weather forecasting, but their performances lagged behind those of numerical models. However, significant improvements have been made in recent years due to better model architectures and training recipes, and increasing data and compute (Keisler, 2022; Pathak et al., 2022; Nguyen et al., 2023; Bi et al., 2023; Lam et al., 2023; Chen et al., 2023a;c). Pangu-Weather (Bi et al., 2023), a 3D Earth-Specific Transformer model trained on $0.25°$ data ($721 \times 1440$ grids), was the first model to outperform operational IFS (Wedi et al., 2015). Shortly after, GraphCast (Lam et al., 2023) scaled up the graph neural network architecture proposed by Keisler (2022) to $0.25°$ data and showed improvements over Pangu-Weather. Despite impressive forecast accuracy, existing methods often involve complex, highly customized neural network architectures with minimal ablation studies, making it difficult to identify which components actually contribute to their success. For example, it is unclear what are the benefits of 3D Earth-Specific Transformer over a standard Transformer, and how critical is the multi-mesh message-passing in GraphCast to its performance. A deeper understanding, and ideally a simplification, of these existing approaches is essential for future progress in the field. Furthermore, establishing a common framework would facilitate the development of foundation models for weather and climate that extend beyond weather forecasting (Nguyen et al., 2023).

In this paper, we show that a simple architecture with a proper training recipe can perform competitively with state-of-the-art methods. We start with a standard vision transformer (ViT) architecture, and through extensive ablation studies, identify the three key components to the performance of the model: (1) a weather-specific embedding layer that transforms the input data to a sequence of tokens by modeling the interactions among atmospheric variables; (2) a randomized dynamics forecasting objective that trains the model to predict the weather dynamics at random intervals; and (3) a pressure-weighted loss that weights variables at different pressure levels in the loss function to approximate the density at each pressure level. During inference, our proposed randomized dynamics forecasting objective allows a single model to produce multiple forecasts for a specified lead time by using different combinations of the intervals for which the model was trained. For example, one can obtain a 3-day forecast by either rolling out the 6-hour predictions 12 times or 12-hour predictions 6 times. Combining these forecasts leads to significant performance improvements, especially for long lead times. We evaluate our proposed method, namely **S**calable **t**ransf**orm**ers for weath**er** forecasting (Stormer), on WeatherBench 2 (Rasp et al., 2023), a widely used benchmark for data-driven weather forecasting. Experiments show that Stormer achieves competitive forecast accuracy of key atmospheric variables for 1–7 days and outperforms the state-of-the-art beyond 7 days. Notably, Stormer achieves this performance by training on more than $5\times$ lower-resolution data and orders-of-magnitude fewer GPU hours compared to the baselines. Finally, our scaling analysis shows that the performance of Stormer improves consistently with increases in model capacity and data size, demonstrating the potential for further improvements.

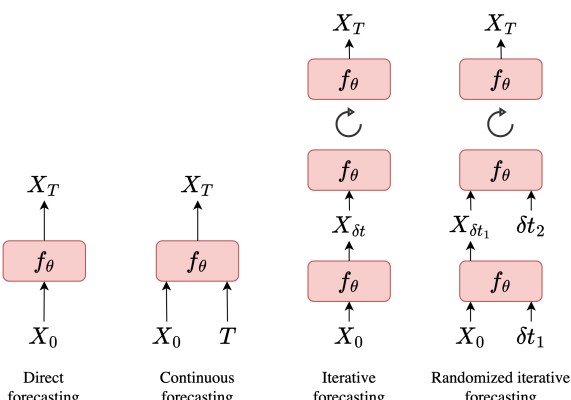

Figure 2: Different approaches to weather forecasting. Direct and continuous methods output forecasts directly, but continuous forecasting is adaptable to various lead times by conditioning on $T$. Iterative forecasting generates forecasts at small intervals $\delta t$, which are rolled out for the final forecast. Our proposed randomized iterative forecasting combines continuous and iterative methods.

## B  BACKGROUND AND PRELIMINARIES

Given a dataset $\mathcal{D} = \{X_i\}_{i=1}^{N}$ of historical weather data, the task of global weather forecasting is to forecast future weather conditions $X_T \in \mathbb{R}^{V \times H \times W}$ given initial conditions $X_0 \in \mathbb{R}^{V \times H \times W}$, in which $T$ is the target lead time, e.g., 7 days; $V$ is the number of input and output atmospheric variables, such as temperature and humidity; and $H \times W$ is the spatial resolution of the data, which depends on how densely we grid the globe. This formulation is similar to many image-to-image tasks in computer vision such as segmentation or video frame prediction. However, unlike the RGB channels in natural images, weather data can contain up to 100s of channels. These channels represent actual physical variables that can be unbounded in values and follow complex laws governed by atmospheric physics. Therefore, the ability to model the spatial and temporal correlations between these variables is crucial to forecasting.

There are three major approaches to data-driven weather forecasting. The first and simplest is *direct forecasting*, which trains the model to directly output future weather $\widehat{X}_T = f_\theta(X_0)$ for each target lead time $T$. Most early works in the field adopt this approach (Dueben & Bauer, 2018; Scher, 2018; Weyn et al., 2019; Rasp & Thuerey, 2021; Clare et al., 2021; Weyn et al., 2020). Since the weather is a chaotic system, forecasting the future directly for large $T$ is challenging, which may explain the poor performances of these early models. Moreover, direct forecasting requires training one neural network for each lead time, which can be computationally expensive when the number of target lead times increases. To avoid the latter issue, *continuous forecasting* uses $T$ as an additional input: $\widehat{X}_T = f_\theta(X_0, T)$, allowing a single model to produce forecasts at any target lead time after training. MetNet (Sønderby et al., 2020; Espeholt et al., 2022; Andrychowicz et al., 2023) employed the continuous approach for nowcasting at different lead times up to 24 hours, WeatherBench (Rasp & Thuerey, 2021) considered continuous forecasting as one of the baselines, and ClimaX (Nguyen et al., 2023) used this approach for pretraining. However, since this approach still attempts to forecast future weather directly, it suffers from the same challenging problem of forecasting the chaotic weather in one step. Finally, *iterative forecasting* trains the model to produce forecasts at a small interval $\widehat{X}_{\delta t} = f_\theta(X_0)$, in which $\delta t$ is typically from 6 to 24 hours. To produce longer-horizon forecasts, we roll out the model by iteratively feeding its predictions back in as input. This is a common paradigm in both traditional NWP systems and the two state-of-the-art deep learning methods, Pangu-Weather and GraphCast. One drawback of this approach is error accumulation when the number of rollout steps increases, which can be mitigated by a multi-step loss function (Keisler, 2022; Lam et al., 2023; Chen et al., 2023a;c). In iterative forecasting, one can forecast either the weather conditions $X_{\delta t}$ or the weather dynamics $\Delta_{\delta t} = X_{\delta t} - X_0$, and $X_{\delta t}$ can be recovered by adding the predicted dynamics to the initial conditions. In this work, we adopt the latter approach, which we refer to as *iterative dynamics forecasting*. We show empirically that our approach achieves superior performance relative to the former approach in Section F.2. Figure 2 summarizes these different approaches.

## C  RELATED WORK

Deep learning offers a promising approach to weather forecasting due to its fast inference and high expressivity. Early efforts (Dueben & Bauer, 2018; Scher, 2018; Weyn et al., 2019) attempted training simple architectures on small weather datasets. To facilitate progress in the field, Weather-Bench (Rasp et al., 2020) provided standard datasets and benchmarks, leading to subsequent works that trained Resnet (He et al., 2016) and UNet architectures (Weyn et al., 2020) for weather forecasting. These works demonstrated the potential of deep learning but still displayed inferior forecast accuracy to numerical systems. However, significant improvements have been made in the last few years. Keisler (2022) proposed a graph neural network (GNN) that performs iterative forecasting with 6-hour intervals, performing comparably with some NWP models. FourCastNet (Pathak et al., 2022) trained an adaptive Fourier neural operator and was the first neural network to run on $0.25°$ data. Pangu-Weather (Bi et al., 2023), with its 3D Earth-Specific Transformer design, trained on high-resolution data, surpassed the benchmark IFS model. Following this, GraphCast (Lam et al., 2023) scaled up Keisler's GNN architecture to $0.25°$, outperforming Pangu-Weather. FuXi (Chen et al., 2023b) was a subsequent work that trained a SwinV2 (Liu et al., 2022) on $0.25°$ data and showed improvements over GraphCast at long lead times. However, FuXi requires finetuning multiple models specialized for different time ranges, increasing model complexity and computation. FengWu (Chen et al., 2023a) was a concurrent work with FuXi that also focused on improving long-horizon forecasts, but has not revealed complete details about their model architecture and training.

## D  DISCUSSION

### D.1  RANDOMIZED FORECASTING

We train Stormer to forecast the dynamics at random intervals $\delta t$ by conditioning on $\delta t$ according to Equation (1). From a practical standpoint, this randomized objective provides two main benefits. First, by randomizing $\delta t$, it enlarges the training data, serving as a form of data augmentation. Second, it enables a single model, once trained, to generate various forecasts for a specified lead time $T$. This is achieved by creating different combinations of the intervals $\delta t$ used in training to sum up to $T$. For instance, to predict weather conditions 7 days ahead, one could roll out the 12-hour forecasts 14 times or the 24-hour forecasts 7 times. Our experiments demonstrate that the amalgamation of these forecasts is crucial for achieving good forecast accuracy, particularly for longer lead times. We note that while both our approach and the continuous approach use the time interval as an additional input to the model, we perform iterative forecasting as opposed to direct forecasting of the counterpart. This avoids the challenge of modeling the chaotic weather directly, as well as offers more flexibility for combining different intervals at test time.

### D.2  INFERENCE

Due to the randomized forecasting objective, Stormer can generate multiple forecasts for a target lead time $T$ by creating different combinations of $\delta t$ that sum to $T$. We consider two inference strategies for generating forecasts, *homogeneous* and *best $m$ in $n$*. The two strategies offer a trade-off between efficiency and expressivity. The homogeneous strategy only requires running three combinations for each lead time $T$, while best $m$ in $n$ provides greater expressivity. Upon determining these combinations and executing the model rollouts, we obtain the final forecast by averaging the individual predictions. This achieves a similar effect to ensembling in NWP, where multiple forecasts are generated by running NWP models with different perturbed versions of the initial condition (Lewis, 2005). As target lead times extend beyond 5–7 days and individual forecasts begin to diverge due to the chaotic nature of the atmosphere, averaging these forecasts is a Monte Carlo integration approach to handle this sensitivity to initial conditions and the uncertainty in the analyses used as initial conditions (Metropolis & Ulam, 1949). We note that our inference strategy is distinguished from that used in Pangu-Weather. While Pangu-Weather trains a separate model for each time interval $\delta t$, we train a single model for all $\delta t$ values by conditioning on $\delta t$. Additionally, while Pangu-Weather relies on a single combination of intervals to minimize rollout steps, our method improves forecast accuracy by averaging multiple forecasts derived from diverse combinations.

### D.3 WEATHER-SPECIFIC EMBEDDING

To capture the complex interactions among input atmospheric variables, we adopt for Stormer a weather-specific embedding module, consisting of two components, *variable tokenization* and *variable aggregation*. Figure 6b compares weather-specific embedding with standard patch embedding, which shows the superior performance of weather-specific embedding at all lead times from 1 to 10 days. A similar weather-specific embedding module was introduced by ClimaX (Nguyen et al., 2023) to improve the model's flexibility when handling diverse data sources with heterogeneous input variables. We show that this specialized embedding module outperforms the standard patch embedding even when trained on a single dataset, due to its ability to effectively model interactions between atmospheric variables through cross-attention.

## E EXPERIMENT DETAILS

### E.1 BASELINES

We compare the forecast performance of Stormer with Pangu-Weather (Bi et al., 2023) and Graph-Cast (Lam et al., 2023), two leading deep learning methods for weather forecasting. Pangu-Weather employs a 3D Earth-Specific Transformer architecture trained on the same variables as Stormer, but with hourly data and a higher spatial resolution of $0.25°$. GraphCast is a graph neural network that was trained on 6-hourly ERA5 data at $0.25°$, using 37 pressure levels for the atmospheric variables, and two additional variables, total precipitation and vertical wind speed. Both Pangu-Weather and GraphCast are iterative methods. GraphCast operates at 6-hour intervals, while Pangu-Weather uses four distinct models for 1-, 3-, 6-, and 24-hour intervals, and combines them to produce forecasts for specific lead times. We include Climatology as a simple baseline. We also compare Stormer with IFS HRES, the state-of-the-art numerical forecasting system, and IFS ENS (mean), which is the ensemble version of IFS. Since WB2 does not provide forecasts of these numerical models beyond 10 days, we defer the comparison against these models to Appendix F.1.

### E.2 STORMER ARCHITECTURE

Figure 3 illustrates the architecture of Stormer. The variable tokenization module tokenizes each variable of the input $X_0 \in \mathbb{R}^{V \times H \times W}$ separately, resulting in a sequence of $V \times (H/p) \times (W/p)$ tokens, where $p$ is the patch size. The variable aggregation module then performs cross-attention over the variable dimension and outputs a sequence of $(H/p) \times (W/p)$ tokens. The interval $\delta t$ is embedded and fed to the Stormer backbone together with the tokens. The output of the last Stormer block is then passed through a linear layer and reshaped to produce the prediction $\Delta_{\delta t}$. Each Stormer block employs adaptive layer normalization to condition on additional information from $\delta t$. Specifically, the scale and shift parameters $(\gamma_1, \beta_1)$ and $(\gamma_2, \beta_2)$ are output by an MLP which takes $\delta t$ embedding as input. This MLP network additionally outputs $\alpha_1$ and $\alpha_2$ to scale the output of the attention and fully connected layers, respectively.

For the main comparison in Section 2, we report the results of our largest Stormer model with 24 transformer blocks, 1024 hidden dimensions, and a patch size of 2, which is equivalent to ViT-L except for the smaller patch size. We vary the model size and patch size in the scaling analysis. For the remaining experiments, we report the performance of the same model as for the main result, but with a larger patch size of 4 for faster training.

In all experiments, the variable tokenization module is a standard patch embedding layer usually used in ViT, and the aggregation module is a single-layer multi-head cross-attention. The first embedding of $\delta t$ is a linear layer, and the adaLN module in each block employs a 2-layer MLP.

For the main comparison with the current methods, we train a Stormer model with a patch size of 2, 1024 hidden dimensions, and 24 Stormer blocks. For the scaling experiments, we vary the hidden dimensions, number of blocks, and patch size. For the rest of the ablation studies, we use a patch size of 4, hidden dimension of 1024, and 24 blocks.

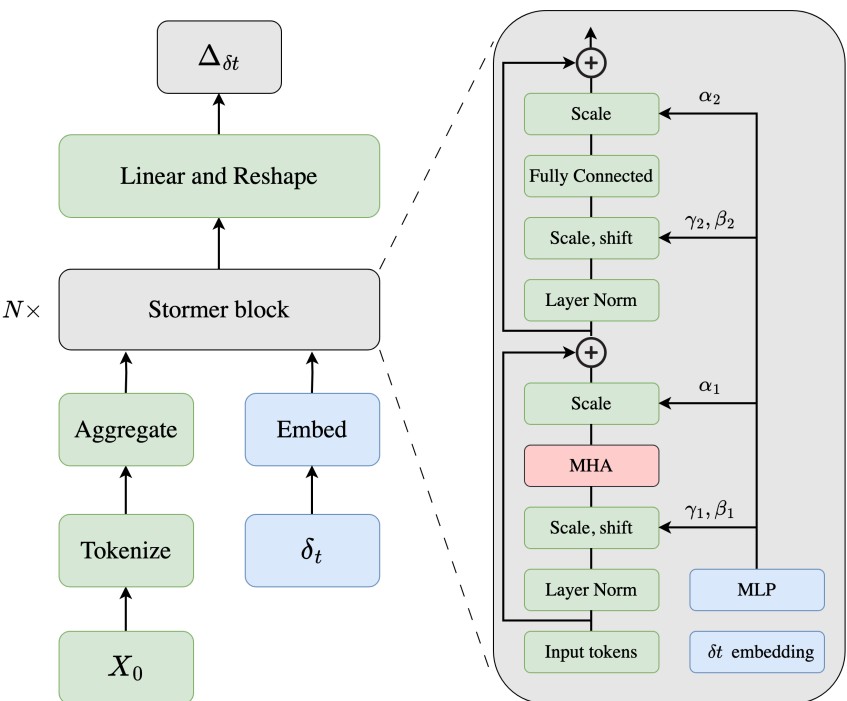

Figure 3: Stormer architecture. The initial condition goes through tokenization and aggregation, before being fed to a stack of $N$ Stormer blocks together with $\delta t$. Each Stormer block employs adaptive layer norm to condition on $\delta t$.

### E.3  Training and evaluation details

#### E.3.1  Data normalization

**Input normalization**  We compute the mean and standard deviation for each variable in the input across all spatial positions and all data points in the training set. This means each variable is associated with a scalar mean and scalar standard deviation. During training, we standardize each variable by subtracting it from the associated mean and dividing it by the standard deviation.

**Output normalization**  Unlike the input, the output that the model learns to predict is the difference between two consecutive steps. Therefore, for each variable, we compute the mean and standard deviation of the difference between two consecutive steps in the training set. What it means to be "consecutive" depends on the time interval $\delta t$. If $\delta t = 6$, we collect all pairs in training data that are 6-hour apart, compute the difference between two data points in each pair, and then compute the mean and standard deviation of these differences. Since we train Stormer with randomized $\delta t$, we repeat the same process for each value of $\delta t$.

#### E.3.2  Three-phase training

As mentioned in Section 1, we train Stormer in three phases with the following objective:

$$\mathcal{L}(\theta) = \mathbb{E}\left[\frac{1}{KVHW}\sum_{k=1}^{K}\sum_{v=1}^{V}\sum_{i=1}^{H}\sum_{j=1}^{W}w(v)L(i)(\widehat{\Delta}_{k\delta t}^{vij} - \Delta_{k\delta t}^{vij})^2\right], \tag{4}$$

where the number of rollout steps $K$ is equal to 1, 4, and 8 in phase 1, 2, and 3, respectively. For phases 2 and 3, we finetune the best checkpoint from the preceding phase.

### E.3.3 PRESSURE WEIGHTS

For pressure-level variables, we assign weights proportionally to the pressure level of each variable. For 4 surface variables, we assign $w = 1$ for T2m and $w = 0.1$ for the remaining variables U10, V10, and MSLP. The surface weights were proposed by GraphCast (Lam et al., 2023) and we did not perform any additional hyperparameter tuning.

### E.3.4 OPTIMIZATION

For the main result in Section 2, we train Stormer in three phases, as described in Section 1.1.2.

For the 1st phase, we train the model for 100 epochs. We optimize the model using AdamW (Kingma & Ba, 2014) with learning rate of $5e - 4$, parameters ($\beta_1 1 = 0.9, \beta_2 = 0.95$) and weight decay of $1e - 5$. We used a linear warmup schedule for 10 epochs, followed by a cosine schedule for 90 epochs.

For the 2nd and 3rd phases, we train the model for 20 epochs with a learning rate of $5e - 6$ and $5e - 7$, respectively. We used a linear warmup schedule for 5 epochs, followed by a cosine schedule for 15 epochs. Other hyperparameters remain the same.

We perform early stopping for all phases, where the criterion is the validation loss aggregated across all variables at lead times of 1 day, 3 days, and 5 days for phases 1, 2, and 3, respectively. We save the best checkpoint for each phase using the same criterion. For the remaining experiments, we only train Stormer for the first phase due to computational constraints.

### E.3.5 SOFTWARE AND HARDWARE STACK

We use PyTorch (Paszke et al., 2019), Pytorch Lightning (Falcon, 2019), timm (Wightman, 2019), numpy (Harris et al., 2020) and xarray (Hoyer & Hamman, 2017) for data processing and model training. We trained Stormer on 128 40GB A100 devices. We leverage mixed-precision training, Fully Sharded Data Parallel, and gradient checkpointing to reduce memory.

### E.3.6 EVALUATION PROTOCOL

We evaluate Stormer and the baselines on forecasting nine key variables: T2m, U10, V10, MSLP, Z500, T850, Q700, U850, and V850. These variables are also used to report the headline scores in WB2. For each variable, we evaluate the forecast accuracy at lead times from 1 to 14 days, using the latitude-weighted root-mean-square error (RMSE) metric. For the main results, we use best $m$ in $n$ inference for rolling out Stormer as it yields the best result, with $m = 32$ and $n = 128$ chosen at random from all possible combinations.

As different models are trained on different resolutions of data, we follow the practice in WB2 to regrid the forecasts of all models to the same resolution of $1.40625°$ ($128 \times 256$ grid points). We then calculate evaluation metrics on this shared resolution. Similarly to WB2, we evaluate forecasts with initial conditions at 00/12UTC for all days in 2020. We provide additional metrics and results of Stormer in Appendix F. For the remaining experiments, we use homogeneous inference for efficiency.

## F ADDITIONAL RESULTS

### F.1 COMPARISON WITH SOTA MODELS

Figure 4 compares Stormer with both deep learning and numerical methods. We take IFS and IFS ENS from WB2 which is only available until day 10. Similar to its deep learning counterparts, Stormer achieves lower RMSE compared to the IFS model for most variables, except for near-surface temperature (T2m) at initial lead times, and only performs slightly worse than IFS ENS. To the best of our knowledge, Stormer is the first model trained on $1.40625°$ data to surpass IFS.

Additionally, we compare Stormer and the baselines on latitude-weighted ACC, another common verification metric for weather forecast models. ACC represents the Pearson correlation coefficient

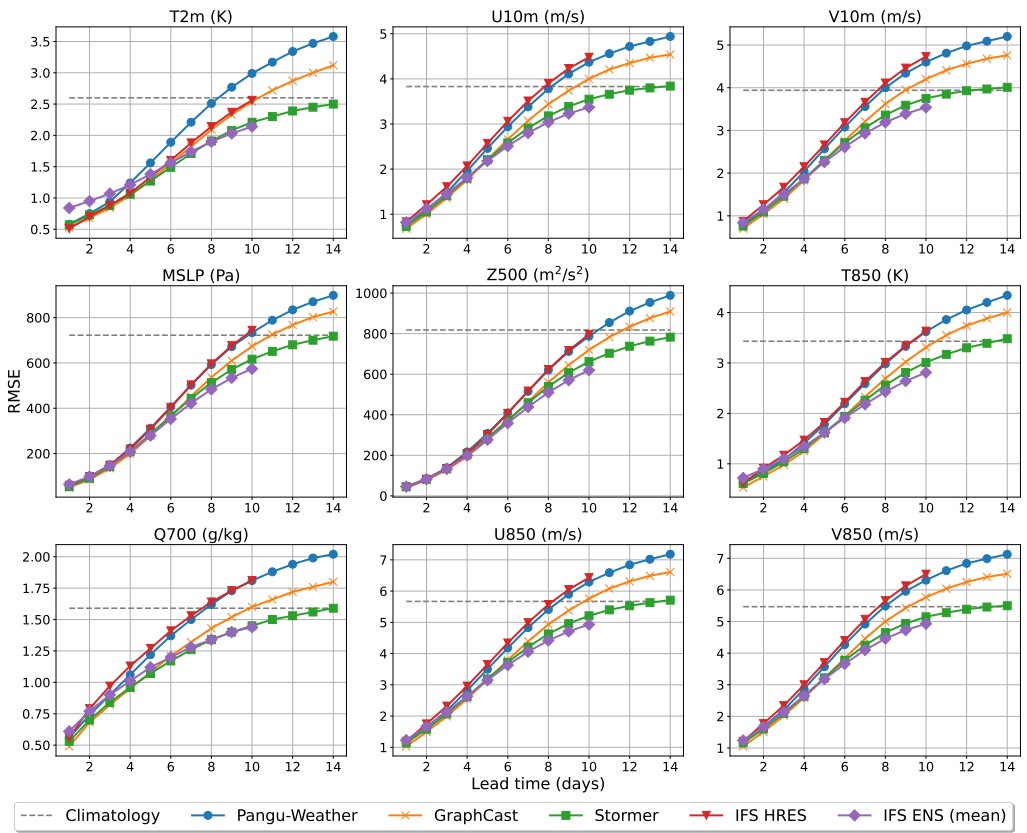

Figure 4: Global forecast verification results of Stormer and the baselines from 1- to 14-day lead times. We show the latitude-weighted RMSE for select variables. Stormer is on par or outperforms each of the benchmark models for the shown variables. During the later portion of the forecasts, Stormer significantly outperforms the current methods.

between forecast anomalies relative to climatology and ground truth anomalies relative to climatology. ACC ranges from −1 to 1, where 1 indicates perfect correlation, and −1 indicates perfect anti-correlation. We refer to WB2 (Rasp et al., 2023) for the formulation of ACC. Figure 5 shows that similarly to RMSE, Stormer achieves competitive performance from 1 to 5 days, and outperforms the baselines by a large margin beyond 6 days.

### F.2 ABLATION STUDIES

We analyze the significance of individual elements within Stormer by systematically omitting one component at a time and observing the difference in performance. First, Figure 6a shows the forecast performance of different single-interval models, which shows that while a small interval works well for short lead times, a larger interval excels at long-term forecasts. This motivates our randomized forecasting objective which trains a model that can produce forecasts at multiple intervals. Figure 6b and 6c demonstrate the importance of weather-specific embedding and adaptive layer normalization (adaLN) to the performance of Stormer.

**Impact of randomized forecasts:** We evaluate the effectiveness of our proposed randomized iterative forecasting approach. Figure 7a compares the forecast accuracy on surface temperature of Stormer and three models trained with different values of $\delta t$. Stormer consistently outperforms all single-interval models at all lead times, and the performance gap widens as the lead time increases. We attribute this result to the ability of Stormer to produce multiple forecasts and combine them to improve accuracy. We note that Stormer achieves this improvement with no computational overhead compared to the single-interval models, as the different models share the same architecture and were trained for the same duration.

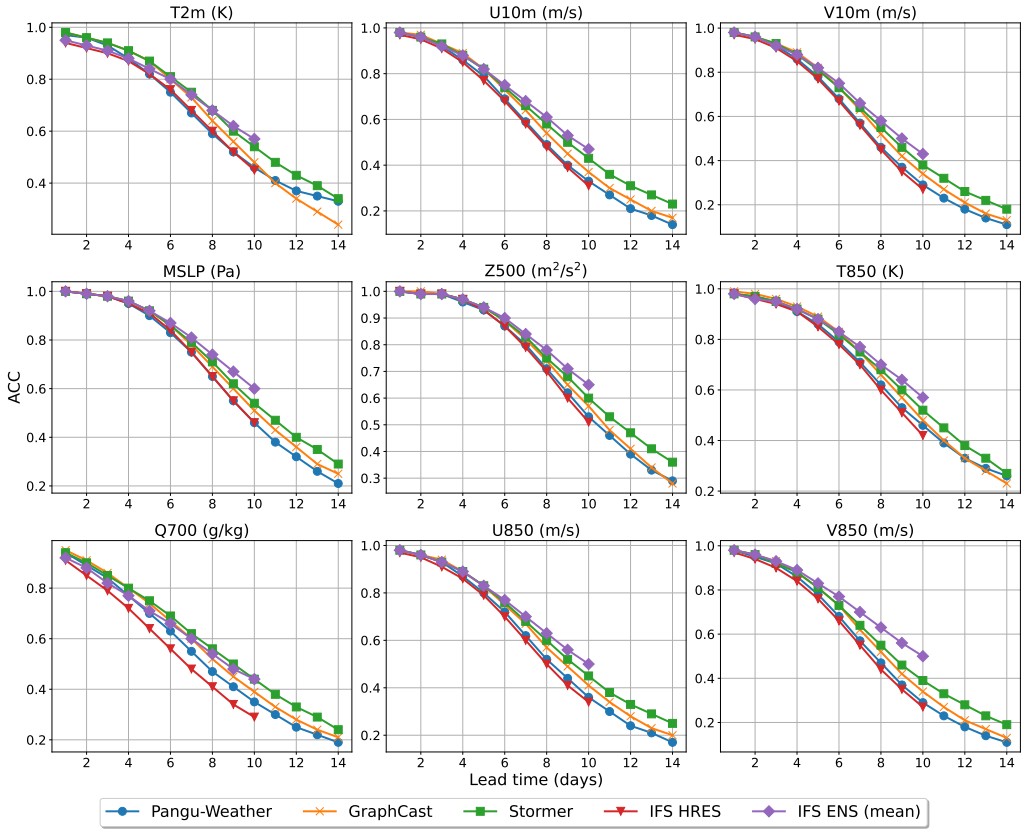

Figure 5: Global forecast verification results of Stormer and the baselines from 1- to 14-day lead times. We show the latitude-weighted ACC for select variables. Stormer is on par or outperforms each of the benchmark models for the shown variables. During the later portion of the forecasts, Stormer significantly outperforms the current methods.

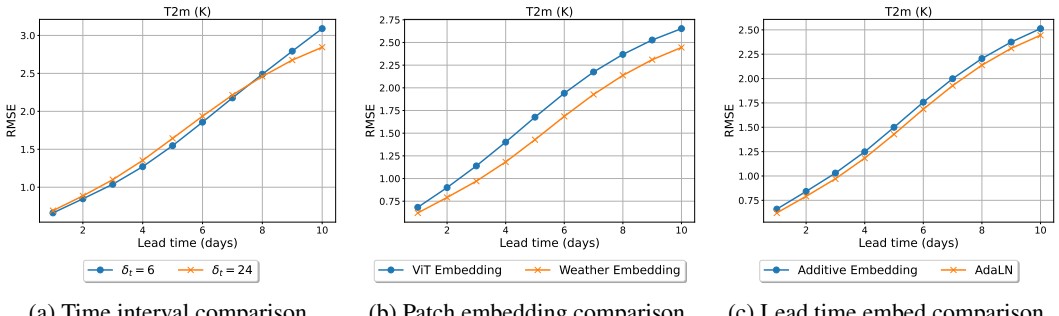

(a) Time interval comparison.  (b) Patch embedding comparison.  (c) Lead time embed comparison.

Figure 6: Preliminary results on surface temperature forecasting that led to the design choices of Stormer: (a) $\delta t = 6$ works well for small lead times, but $\delta t = 24$ excels at lead times beyond 7 days. (b) Weather-specific embedding significantly outperforms standard ViT embedding. (c) Adaptive layer norm outperforms additive embedding. Similar trends are observed across different output variables.

**Impact of pressure-weighted loss:** Figure 7b shows the superior performance of Stormer when trained with the pressure-weighted loss. Intuitively, the weighting factor prioritizes variables that are nearer to the surface, as these variables are more important for weather forecasting and climate science. The pressure-weighted loss was first introduced by GraphCast, and we show that it also helps with a different architecture.

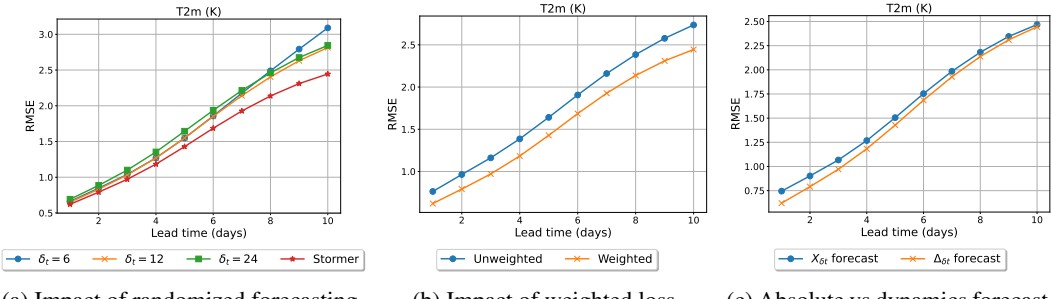

(a) Impact of randomized forecasting.  (b) Impact of weighted loss.  (c) Absolute vs dynamics forecast.

Figure 7: Ablation studies showing the importance of each component in Stormer: (a) Stormer outperforms single-interval models at all lead times. (b) Pressure-weighted loss improves accuracy significantly. (c) Dynamics forecasting is consistently better than absolute forecast. Similar trends are observed across different output variables.

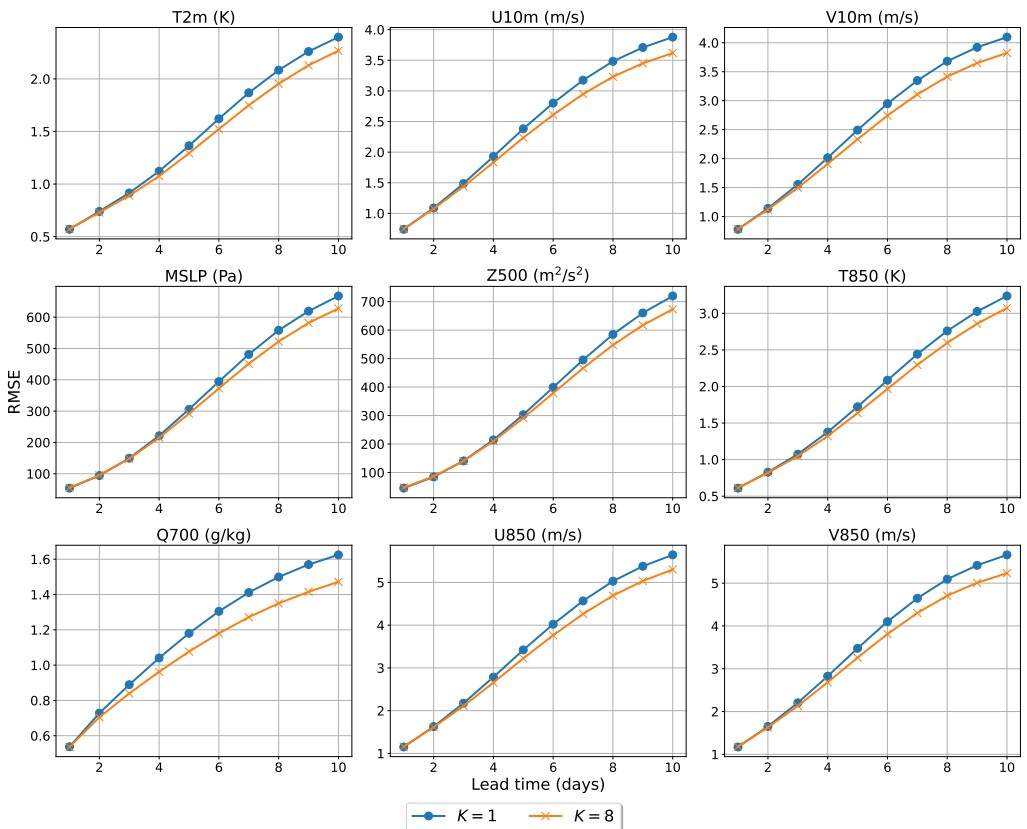

Figure 8: Performance of Stormer without ($K = 1$) and with ($K = 8$) multi-step fine-tuning.

**Dynamics vs. absolute forecasts:** We justify our decision to forecast the dynamics $\Delta_{\delta t}$ by comparing with a counterpart that forecasts $X_{\delta t}$. Figure 7c shows that forecasting the changes in weather conditions (dynamics) is consistently more accurate than predicting complete weather states. One possible explanation for this result is that it is simpler for the model to predict the changes between two consecutive weather conditions than the entire state of the weather; thus, the model can focus on learning the most significant signal, enhancing forecast accuracy.

**Impact of multi-step fine-tuning** We verify the importance of multi-step fine-tuning by comparing Stormer after the 1st phase ($K = 1$) and after the 3rd phase ($K = 8$). Figure 8 shows that multi-step fine-tuning significantly improves performance at long lead times.

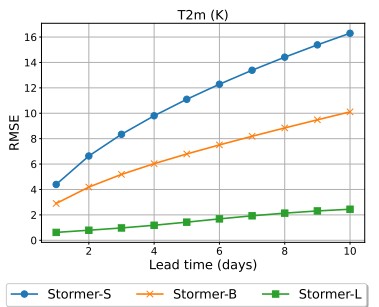 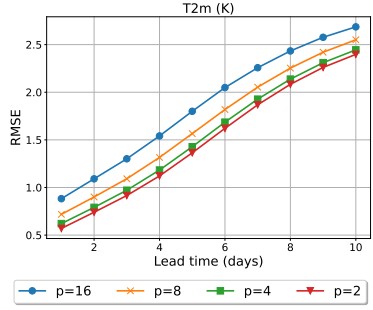

Figure 9: Stormer improves consistently with larger models (left) and smaller patch sizes (right).

## F.3 SCALING ANALYSIS

We examine the scalability of Stormer with respect to model size and the number of training tokens. We evaluate three variants of Stormer – Stormer-S, Stormer-B, and Stormer-L, whose parameter counts are similar to ViT-S, ViT-B, and ViT-L, respectively. To understand the impact of training token count, we vary the patch size from 2 to 16. The number of training tokens increases fourfold whenever the patch size is halved. Figure 9 shows a significant improvement in forecast accuracy when we increase the model size, and the performance gap widens as we increase the lead time. Since we do not perform multi-step fine-tuning for these models, minor performance differences at short intervals may become magnified over time. While multi-step fine-tuning could potentially reduce this gap, it is unlikely to eliminate it entirely. Reducing the patch size also improves the performance of the model consistently. From a practical view, smaller patches mean more tokens and consequently more training data. From a climate perspective, smaller patches capture finer weather details and processes not evident in larger patches, allowing the model to more effectively capture physical dynamics that drive weather patterns.

## F.4 QUALITATIVE RESULTS

We visualize forecasts produced by Stormer at lead times from 1 days to 14 days for 9 key variables. All forecasts are initialized at 0UTC January 26th 2020. Each figure illustrates one lead time, where each row is for each variable. The first column shows the initial condition, the second column shows the ground truth at that lead time, the third column shows the forecast, and the last column shows the bias, which is the difference between the forecast and the ground truth.

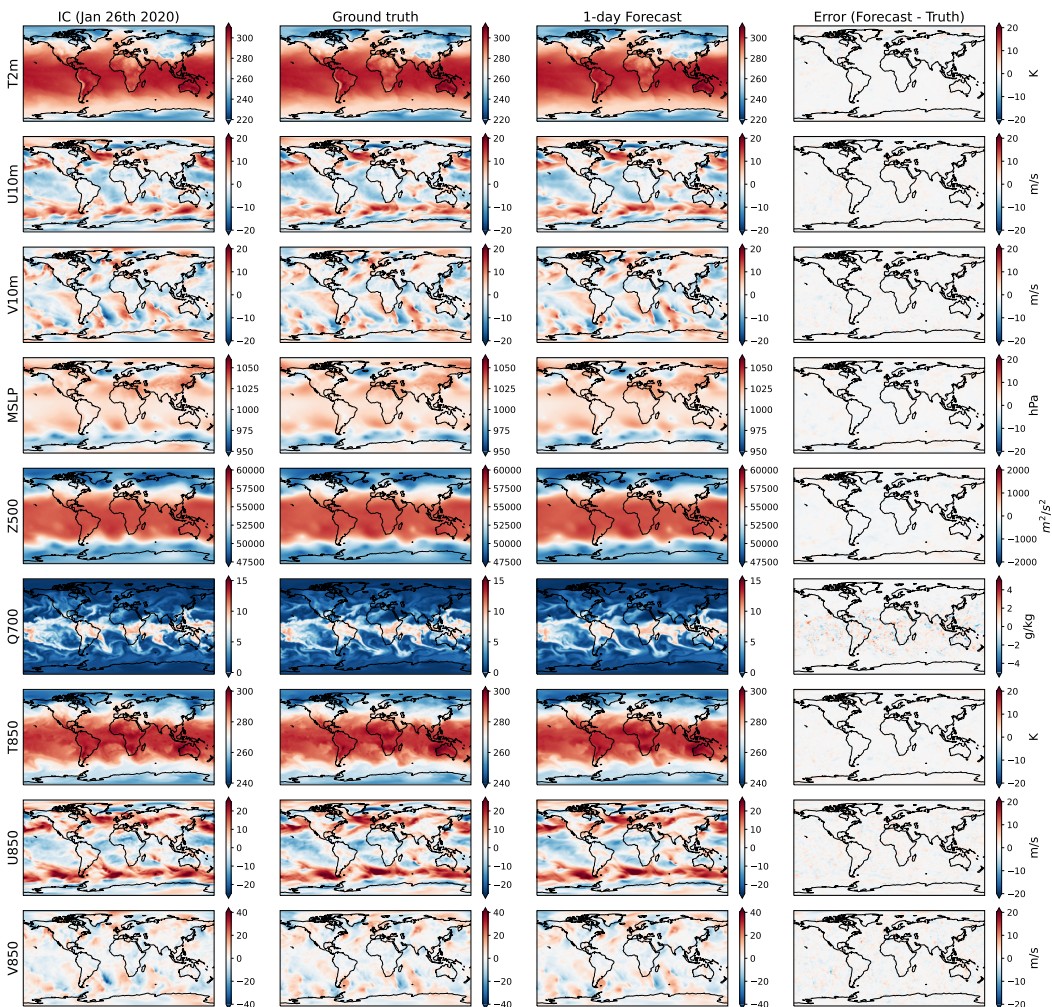

Figure 10: 1-day lead time

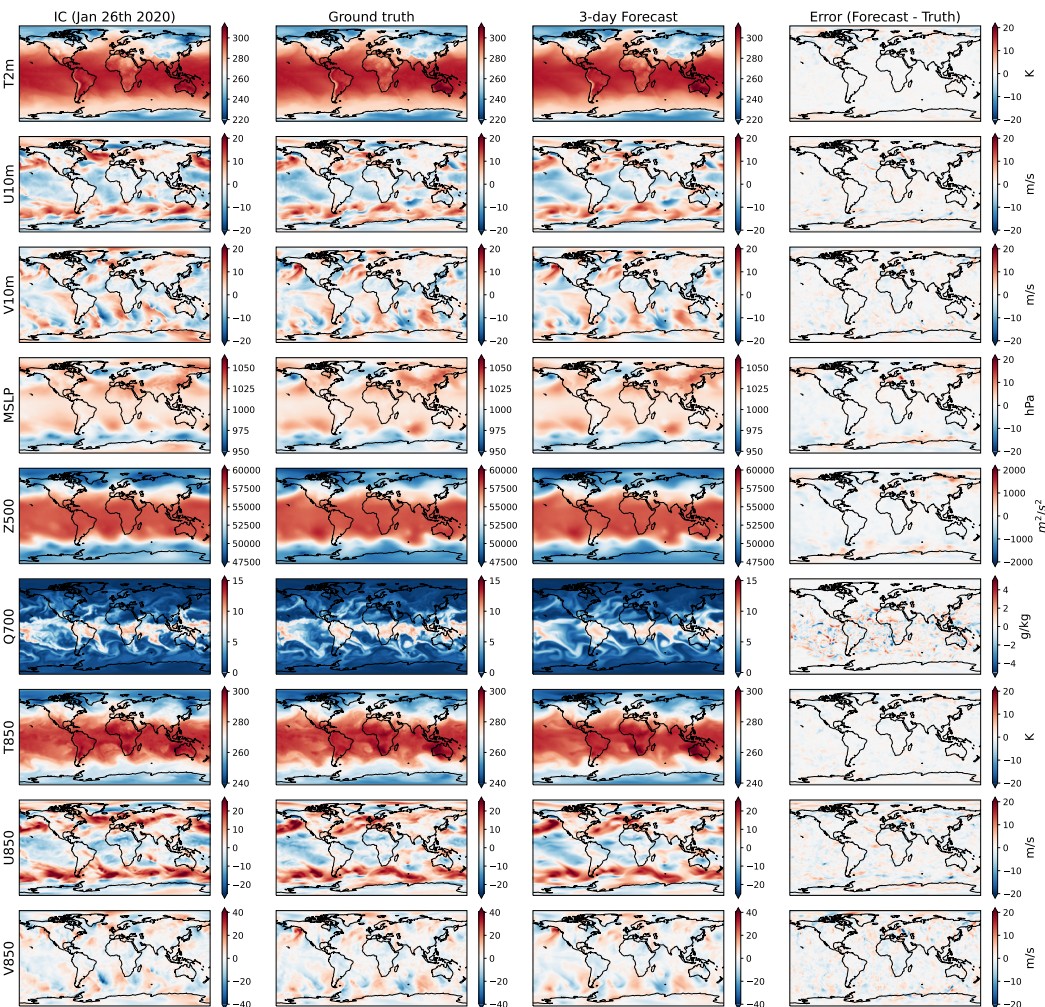

Figure 11: 3-day lead time

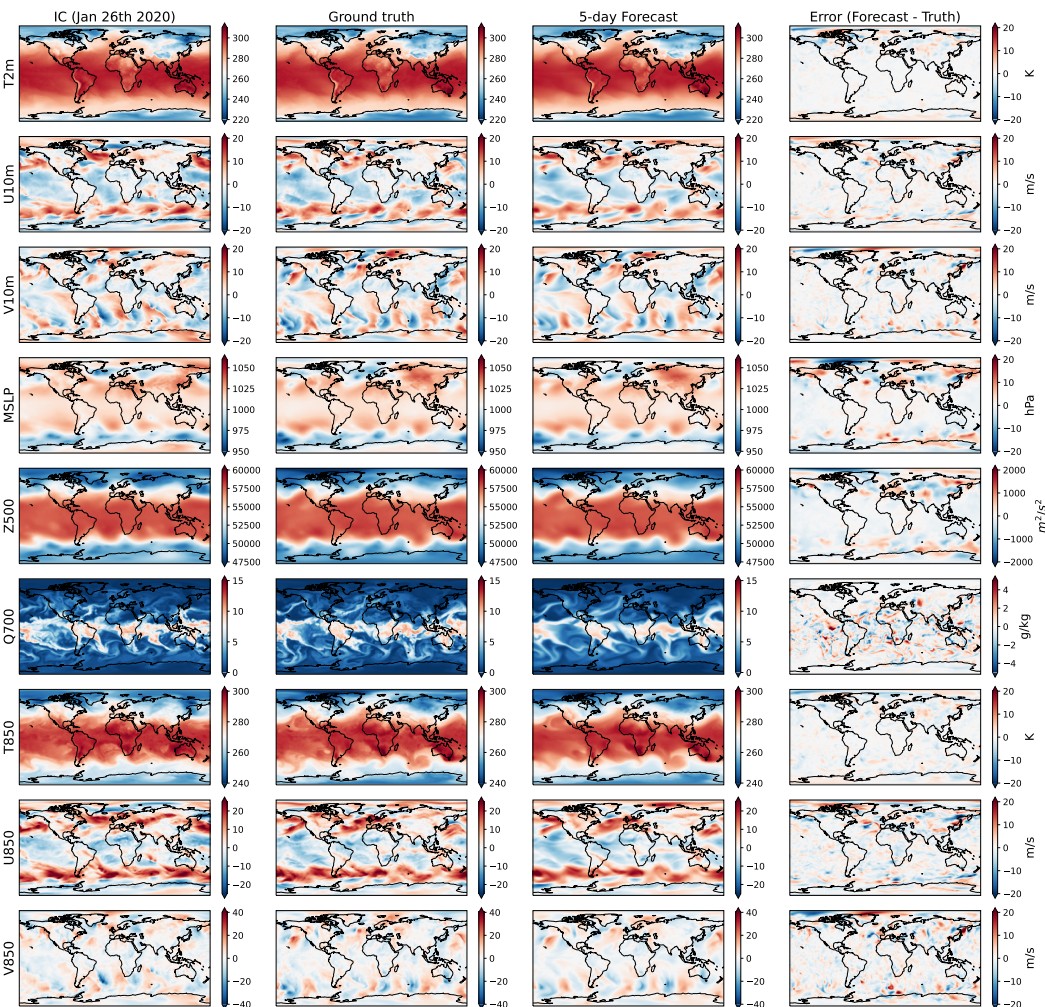

Figure 12: 5-day lead time

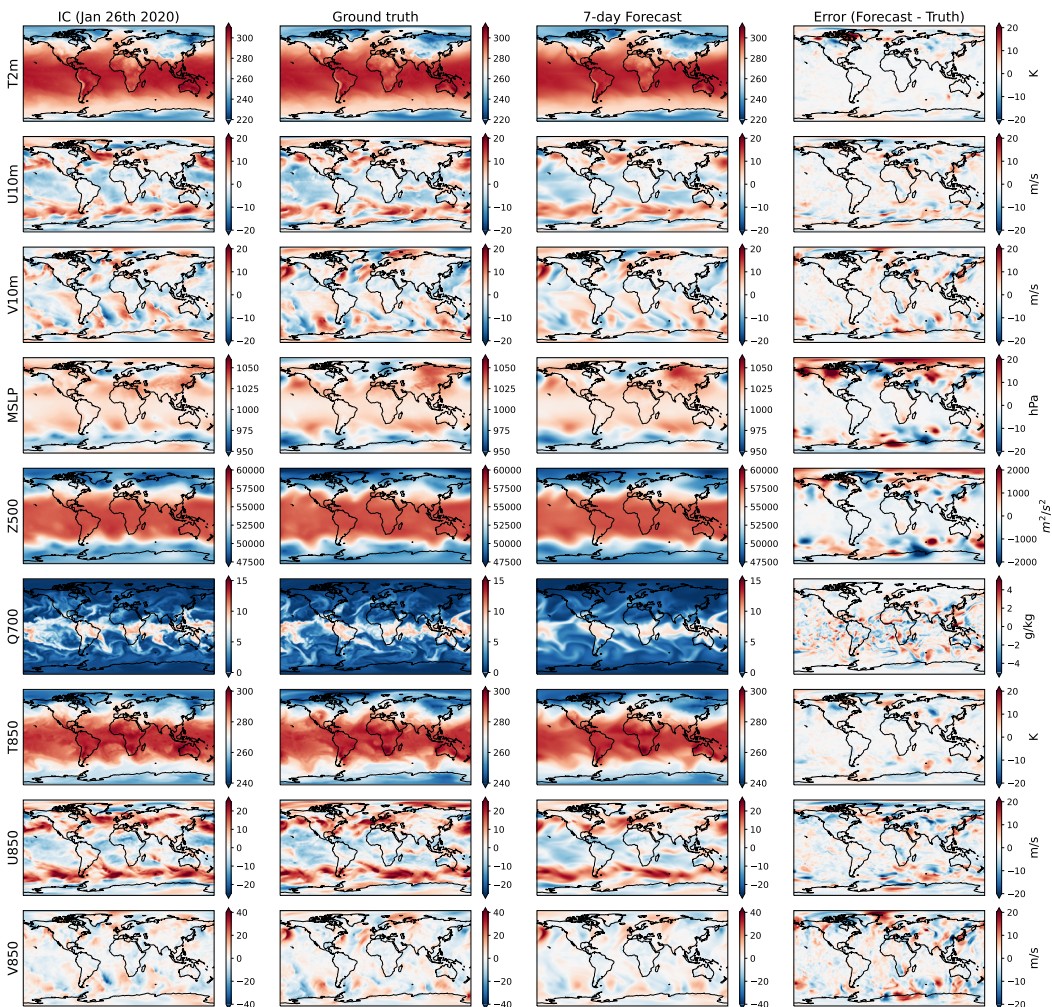

Figure 13: 7-day lead time

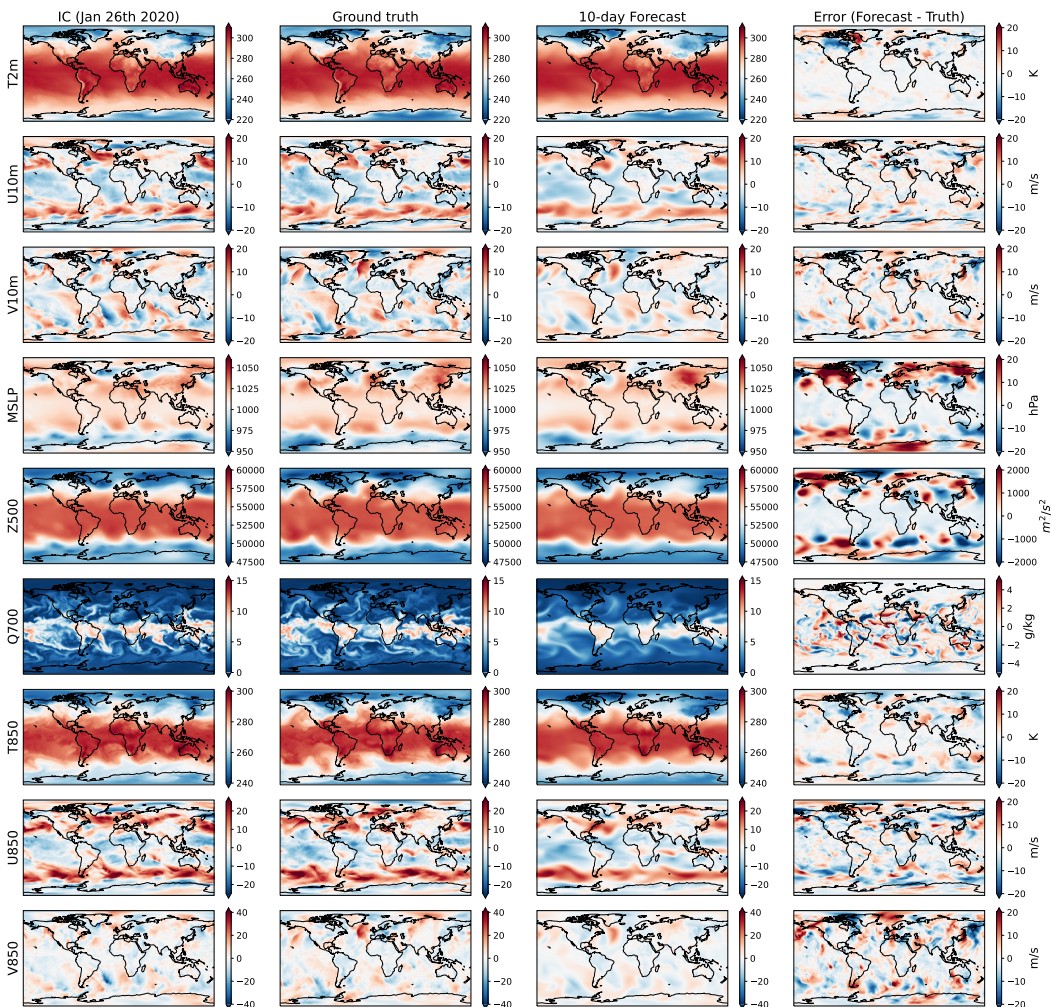

Figure 14: 10-day lead time

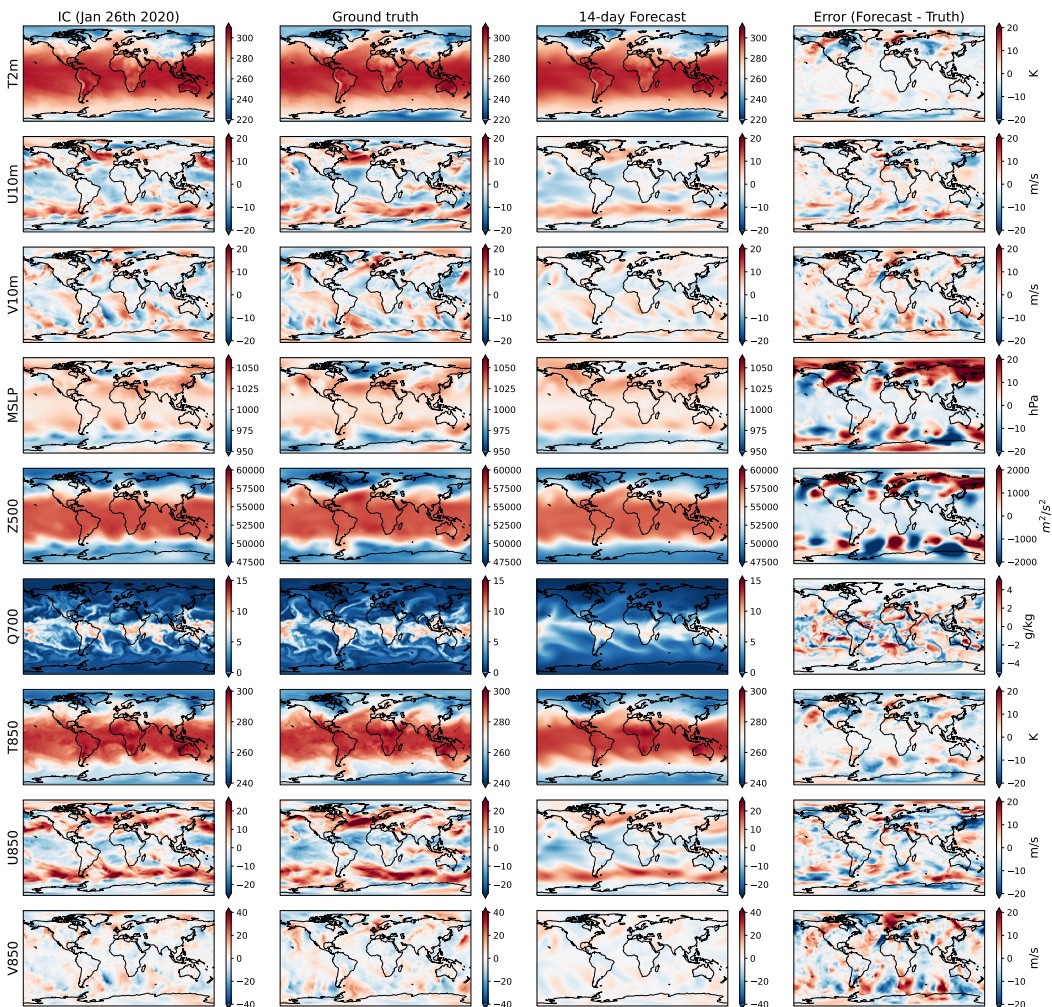

Figure 15: 14-day lead time

