# OpenReview forum: "Scaling Transformers for Skillful and Reliable Medium-range Weather Forecasting"
_ICLR.cc/2024/Workshop/AI4DiffEqtnsInSci — AI4DiffEqtnsInSci @ ICLR 2024 Poster_

### Official Review · Reviewer_RHTZ · 2024-02-26
**Vision transformers for weather forecasting that show SOTA performance with minimal architecture changes**

**Rating:** 6
**Confidence:** 4

**Review:**

The authors present a vision transformer with minimal additions as well as training recipes that make their model competitive with all SOTA models (and even better at certain lead times) for the problem of weather forecasting.
 Strengths:
1. The paper is clear, well-written and presents some contributions that are novel.
2. The paper could have significant impact as they adapt the philosophy of CV/NLP and scale transformers for the weather domain without major changes to the transformer backbone. This can be very useful to researchers in this domain and to obtain scaling laws.
3. The model outperforms SOTA graphcast and pangu models as well as numerical PDE baseline IFS in RMSE/ACC metrics.
4. The paper presents ablations on the modifications of ViT to show their improvements and also demonstrate some minimal scaling laws w.r.t patch size and #parameters of the ViT.

Weaknesses and questions:
1. Loss weighting, multistep finetuning are already present in SOTA models such as graphcast. It will be useful if the authors clearly laid out their contributions in bullet points and for others appropriately referenced where they are from in their sections.
2. The variable aggregation seems to add a lot of memory pressure. For the cross-attention, I would need to store local_batch * V * h/p * w/p * D which is a lot and can easily become bottlenecks at higher resolution. Please comment on how this method will scale.
3. Further, for 2. you add more parameters as well as compute as well. I'm not sure how fair then the comparison is in Fig 6b. Can the authors comment on time per epoch for ViT with weather embedding and ViT without weather embedding assuming everything else stays exactly the same? Same for mem consumed. This would help the reader gauge how scalable the model is and what the cost associated with the weather embedding is -- I assume it doesn't come for free.
4. The compute comparison is between different resolutions. For ex: the authors say pangu takes 15 days to train a model. But they operate on 32x (721/128 * 1440/256) finer data. By comparison, stormer takes 1 day. With 32x finer data, the model would take 1000 days assuming quadratic complexity of the ViT. I think this should be acknowledged -- because a researcher can, in principle, train pangu/graphcast on lowres data and get away with lesser compute.
5. Relating to 4, I'm wondering what the effect of resolution is on RMSE? While it is averaged spatially, coarser fields have lesser fine-scale structure and hence it is possible to get better RMSEs. Have the authors looked at spectra? This also might be useful to motivate Fig 9 (right) as smaller patch sizes may give better spectal metrics.

---

### Official Review · Reviewer_YFRS · 2024-02-27
**An excellent study of key design choices in deep learning weather prediction.**

**Rating:** 9
**Confidence:** 5

**Review:**

Summary: The authors investigate best practices for designing Transformer architectures for weather forecasting. They identify key design choices in existing methods to build a streamlined architecture, with thorough ablation analysis to verify the benefits of their approach.

Pros:
- Strong performance on WeatherBench2 indicates that this is a state-of-the-art model in a competitive environment.
- Identifies key design choices in deep weather prediction models
- Analysis of scaling properties
- Clear and easy to follow methodology and presentation

Cons:
- Further analysis of spectral properties and temporal consistency would be very helpful to gauge the efficacy of the randomized forecasting objective.
- Further analysis of the costs/benefits of training and evaluating on different resolutions than the baselines would make the results more convincing.
- A comparison with more recent models like FuXi would be very interesting!

---

### Meta-Review · Area_Chair_HPXz · 2024-03-01

**Recommendation:** Accept (Poster)

**Metareview:**

This paper presents Stormer for medium range weather forecasting. On WeatherBench 2, Stormer performs competitively at short to medium-range forecasts and outperforms current methods beyond 7 days. The concerns of clarification raised by the reviewers are valid. I strongly command the author to address those concerns in the camera ready version.

---

### Decision · Program_Chairs · 2024-03-01

Accept (Poster)